# The Influence of Salinity on the Removal of Ni and Zn by Sorption onto Iron Oxide- and Manganese Oxide-Coated Sand

**Jiyeon Choi, Ardie Septian and Won Sik Shin ***

School of Architecture, Civil, Environmental, and Energy Engineering, Kyungpook National University, Daegu 41566, Korea; iamchoig@gmail.com (J.C.); ardieseptian@knu.ac.kr (A.S.)

* Correspondence: wshin@knu.ac.kr; Tel.: +82-53-950-7584

**Abstract:** The influence of salinity on the single and binary sorption of Ni and Zn onto iron oxide- and manganese oxide-coated sand (IOCS and MOCS) was investigated at pH = 5. The single sorption experimental data were fitted to Freundlich, Langmuir, Dubinin–Radushkevich, and Sips models, and a nonlinear sorption isotherm was observed ($N_F$ = 0.309–0.567). The higher Brunauer–Emmett–Teller (BET) surface area ($A_{BET}$) and cation exchange capacity (CEC) of MOCS contributed to the higher maximum sorption capacities ($q_{mL}$) of Ni and Zn than that of IOCS. The Ni sorption capacities in the single sorption were higher than that in the binary sorption, while the Zn sorption capacities in the single sorption were less than that in the binary sorption. The single and binary sorptions onto both IOCS and MOCS were affected by the salinity, as indicated by the decrease in sorption capacities. Satisfactory predictions were shown by the binary sorption model fitting including P-factor, ideal adsorbed solution theory (IAST)–Freundlich, IAST–Langmuir, and IAST–Sips; among these, the P-factor model showed the best fitting results in predicting the influence of salinity of Ni and Zn in the binary sorption system onto IOCS and MOCS. IOCS and MOCS offer a sustainable reactive media in a permeable reactive barrier (PRB) for removing Ni and Zn in the presence of salinity.

**Keywords:** binary sorption; heavy metals; IOCS; MOCS; salinity

---

## 1. Introduction

Heavy metal-contaminated groundwater has become an important issue in the coastal regions where freshwater resources are limited [1]. Most of the shallow groundwater resources in the crowded urban areas near coastal regions, such as megacities, were heavily contaminated by heavy metals due to anthropogenic activities [2]. Ni and Zn are frequently found heavy metals that are strongly correlated with groundwater toxicity and coastal sediment, posing serious risks to human health and ecosystems at large [2,3]. The health risks of Ni and Zn are categorized as high based on the cancer risk and hazard quotients evaluation above the Ni and Zn threshold concentration (>3000 $\mu$g L$^{-1}$) [1,4,5]. Recently, Tian et al. [6] and Liu et al. [7] reported the significant contribution of industrialization to the increase in Ni and Zn contaminations in water sources in China and Korea, two heavily industrialized countries where 70–75% of major cities and industries are located in the coastal regions. Attempts to treat Ni and Zn contamination in coastal groundwater are crucial since the economic growth of these two countries, especially Korea, is highly dependent on industrial development alongside the coastal area. Ni and Zn removal from water and wastewater has been reported using various sorbents, such as natural and modified zeolite, bentonite, and vermiculite through adsorption and ion exchange mechanisms [8], although their removal from the groundwater near coastal regions through sorption has not been examined.

Metal oxide-coated sands have been used for removing various heavy metals from water via the sorption mechanism [9]. For example, iron oxides are well-known to give a powerful effect on heavy metal sorption [10]. Due to their high affinities toward metal cations, manganese oxides are also commonly employed to remove heavy metals in storm water infiltration [11]. These metal oxides were usually coated onto a granular solid support such as sand to avoid difficulties in separation from the water after heavy metal sorption treatment, resulting in easy separation through centrifugation or filtration [12]. Iron oxide- and manganese oxide-coated sand (IOCS and MOCS) have been used as permeable reactive barrier (PRB) materials for treating heavy metal contamination in groundwater due to their ability to sorb various heavy metals, low-cost installation, and sustainable characteristics, replaceability, and long life-time [13]. Several studies reported that IOCS was satisfactory in removing Pb(II) [10], Cr(VI) and As(VI) [14], As(III) [15], Mo(VI) [16], and Cu(II) [17], whereas MOCS was effective in removing Cu(II) [9], Cr(VI), Pb(II), and Cd [12], and As(III) and As(VI) [18]. However, few studies have investigated the sorption of Ni and Zn onto IOCS and MOCS [11,17,19].

This study is focused on the treatment of groundwater contaminated by Ni and Zn, especially in the steel industry located in coastal regions. In these regions, the groundwater sources have always faced seawater intrusion, causing the deterioration of groundwater quality through salinization [1]. Moreover, the presence of heavy metals in the binary sorption system brings another challenge. Thus, it is necessary to study the effect of salinity on the binary sorption of Ni and Zn onto IOCS and MOCS as sorbents.

The aim of this study was to investigate the effect of salinity on Ni and Zn single and binary sorption onto IOCS and MOCS. The single sorption data were analyzed using Freundlich, Langmuir, Dubinin Radushkevich (DR), and Sips models. To the best of our knowledge, this is the first report on the effect of salinity on the binary sorption of Ni and Zn onto IOCS and MOCS by applying binary sorption models such as the P-factor model and ideal adsorbed solution theory (IAST). The physicochemical properties of the sorbents were correlated with the single and binary sorption parameters to determine the feasibility of using IOCS and MOCS as permeable reactive barrier (PRB) materials.

## 2. Materials and Methods

### 2.1. Materials

Nickel nitrate hexahydrate ($Ni(NO_3)_2 \cdot 6H_2O$, ≥98%, Sigma–Aldrich, St. Louis, MO, USA) and zinc nitrate hexahydrate ($Zn(NO_3)_2 \cdot 6H_2O$, ≥98%, Kanto Chemical Co., Tokyo, Japan) were used to prepare the aqueous solutions of Ni and Zn, respectively. The ionic strength of Ni and Zn was maintained using 0.01 M $NaNO_3$ (≥99%, Sigma–Aldrich, USA). In all experiments, the chemical reagents (ACS grade) were dissolved using distilled and deionized (DDI) water produced using a MilliporeSigma™ Synergy™ Ultrapure Water Purification System (Thermo Fisher Scientific, Waltham, MA, USA). Artificial seawater was prepared according to a standard method [20]. The artificial seawater consisted of $Na^+$, $K^+$, $Ca^{2+}$, $Mg^{2+}$, $Cl^-$, $SO_4^{2-}$, and $HCO_3^-$, was prepared by dissolving NaCl, KCl, $CaCl_2$, $MgCl_2$, $MgSO_4$, and $NaHCO_3$ in 1 L of DDI water, as shown in Supplementary Materials (SI) Table S1. The pH of artificial seawater was maintained at 5 using a 0.05 M 2-(*N*-morpholino) ethanesulfonic acid (MES, 99.5%, ACROS Organics, Fair Lawn, NJ, USA) buffer solution.

Joomoonjin sand purchased from a local supplier in Korea with a particle size of 1.0–1.2 mm was used as the supporting material for iron oxide and manganese oxide. The sand grain was immersed in 2.0% $H_2O_2$ (30%, Sigma-Aldrich, USA) and washed using DDI water followed by HCl (35–37%, Duksan Pure Chemicals Co., Ltd., Ansan-si, Korea) to remove any impurities on the sand surface.

### 2.2. IOCS and MOCS Preparation

IOCS was prepared by following the method by Boujelben et al. [19], whereas MOCS was synthesized by modifying the methods proposed by Chaudhry et al. [12] and Boujelben et al. [19]. Briefly, 28 mM Fe(III) and Mn(II) stock solutions were prepared by dissolving $FeCl_3.6H_2O$

(97%, Duksan Pure Chemicals Co., Ltd., Korea) and $MnCl_2$ (98%, Duksan Pure Chemicals Co., Ltd., Korea), respectively. For IOCS, 15 g washed sand was added into 250 mL Fe(III) stock solution in a 500 mL beaker, and they were stirred using a mechanical stirrer (Corning$^{TM}$, PSC-420D, Thermo Fischer Scientific, Waltham, MA, USA) and a Teflon-coated magnetic bar at 200 rpm. Then, 0.1 M NaOH was added dropwise into the mixture to obtain a final pH of ~9. The MOCS was prepared in the same manner as that for IOCS by adding into the Mn(II) stock solution followed by the dropwise addition of 200 mL 0.05 M $KMnO_4$ (99.5%, Daejung Co., Busan, Korea). During the preparation of IOCS and MOCS, the solution was continuously stirred at 60–65 °C for 24 h to maintain the oxide growth on the sand surface. The final product was aged for 10 h, filtered, and washed using DDI water. The clean IOCS and MOCS were dried in an oven for 2 d at 60 °C, ground, and sieved through a 212 μm sieve (US standard mesh).

## 2.3. IOCS and MOCS Characterization

The BET surface areas ($A_{BET}$) of IOCS and MOCS were calculated using $N_2$ sorption–desorption data and fitted to the Brunauer–Emmett–Teller (BET) model (Quantachrome, Autosorb-iQ and Quadrasorb Si, Boynton Beach, FL, USA). The sodium acetate method [21] was used to determine the cation exchange capacity (CEC). The point of zero charge ($pH_{PZC}$) was determined using Appel et al.'s method [22]. The pH change was measured by using a pH meter (Orion 3 Star pH Benchtop, Thermo Scientific, Waltham, MA, USA). An X-ray diffractometer (XRD, X'pert PRO MRD, PANalytical, Almelo, The Netherlands) with a CuKα source (λ = 1.54 Å) and a PIXcel$^{3D}$ detection system was used to determine the crystal size. A scanning electron microscope (SEM, S-4200, Hitachi, Chiyoda City, Tokyo, Japan) with energy-dispersive X-ray spectrometry (EDS, Horiba, E-MAX EDS detector, Kyoto, Japan) was used to analyze the morphology and elemental composition.

## 2.4. Sorption Experiments

Prior to the experiments, the IOCS and MOCS as sorbents were rinsed several times with a 0.05 M MES buffer solution to adjust the pH of the sorbents. Series of Ni (0.02–1.70 mmol $L^{-1}$) and Zn solution (0.02–1.53 mmol $L^{-1}$) were prepared using DDI water (0‰) and artificial seawater (30‰) to investigate the influence of salinity on the sorption. The pH of Ni and Zn stock solutions were also adjusted to 5 using the same buffer solution. This pH was chosen to avoid the Ni and Zn precipitation, as shown in Figure S1, where the real groundwater pH near coastal areas was in the range of 4.71–7.53 [23].

The single sorption experiments were conducted at 25 °C using 50 mL polycarbonate vials (Nalgene Co., USA). At first, the polycarbonate vials filled with 1.0 g of sorbents were mixed with eight different initial concentrations of Ni or Zn solution at 1:40 solid-to-liquid ratio (*wt/wt*) without headspace to prevent the effect of carbon dioxide in the air. The vials containing samples were horizontally shaken at 200 rpm for 24 h to reach equilibrium, centrifuged at 3000 rpm (1000× *g*) for 20 min, and filtered using a 0.2 μm syringe filter (cellulose nitrate membrane, φ = 25 mm, Whatman, Darmstadt, Germany). The Ni and Zn in the aqueous phase were analyzed by inductively coupled plasma–optical emission spectroscopy (ICP-OES, Optima 2100DV, PerkinElmer Co., Waltham, MA, USA). By assuming that the concentration changes in the solution phase were caused by the sorption onto the solid phase, the mass balance calculation was used to estimate the heavy metal concentration in the solid phase. All experiments were performed in duplicate.

The binary sorption (Ni/Zn) experiments were performed in the same manner as the single sorption experiments with equimolar metal solutions (0.02–1.70 mmol $L^{-1}$) in a 1:1 volume ratio.

## 2.5. Single Sorption Models

By considering the heterogeneous sorbent surface, the Freundlich model was chosen to fit the single sorption data:

$$q = K_F C^{N_F} \tag{1}$$

where $q$ (mmol g$^{-1}$) and $C$ (mmol L$^{-1}$) are the solid and aqueous-phase equilibrium concentration, respectively, $K_F$ [(mmol g$^{-1}$)/(mmol L$^{-1}$) $^{N_F}$] is the Freundlich sorption coefficient-related sorption affinity, and $N_F$ (–) is the Freundlich coefficient related to nonlinearity. $N_F \neq 1$ defines nonlinear sorption [24].

The Langmuir model is useful to fit the single sorption data when monolayer sorption occurs on a sorbent surface containing a finite number of binding sites. The Langmuir model is represented as

$$q = \frac{q_{mL}b_L C}{1 + b_L C} \tag{2}$$

where $q_{mL}$ (mmol g$^{-1}$) and $b_L$ (L mmol$^{-1}$) are the Langmuir parameters that represent maximum sorption capacity and site energy factor, respectively. This model can also be used to predict whether a sorption system is favorable. The essential feature of the Langmuir isotherm is the dimensionless constant separation factor ($R_L$):

$$R_L = \frac{1}{1 + b_L C_0} \tag{3}$$

where $C_0$ (mmol L$^{-1}$) is the initial metal concentration. The $R_L$ values (–) indicate the type of isotherm to be favorable ($0 < R_L < 1$), or irreversible ($R_L = 0$; [25]).

The DR model is used to identify whether the sorption mechanism occurs through a physical or chemical process [24]:

$$q = q_{mD} \exp(-\beta \varepsilon^2) = q_{mD} \exp\left[-\beta\left(\text{RT } \ln\left(1 + \frac{1}{C}\right)\right)^2\right] \tag{4}$$

where $q_{mD}$ (mmol g$^{-1}$) is the theoretical saturation capacity, $\beta$ is a constant related to the mean free energy of sorption per mole of the sorbate (mol$^2$ J$^{-2}$), and $\varepsilon$ is the Polanyi potential, which is equal to RT ln(1 + 1/C), where $R$ (J mol$^{-1}$ K$^{-1}$) is the gas constant and $T$ (K) is the absolute temperature. $\beta$ indicates the mean free energy $E$ (J mol$^{-1}$) of sorption per molecule of sorbate and can be calculated using the following formula [24]:

$$E = \frac{1}{(2\beta)^{1/2}} \tag{5}$$

where $E$ is used to differentiate the sorption mechanism. The value between 8 and 16 kJ mol$^{-1}$ indicates the domination of the ion exchange mechanism, while the $E$ value < 8 kJ mol$^{-1}$ indicates physical force and > 16 kJ mol$^{-1}$, the chemical process as the dominant sorption mechanisms [25].

The Sips model is a combination of the Freundlich and Langmuir isotherms and can be used to represent the equilibrium sorption data [26,27]:

$$q = \frac{q_{mS}(b_S C)^{N_S}}{1 + (b_S C)^{N_S}} \tag{6}$$

where $q_{mS}$ (mmol/g), $b_S$ (L/mmol), and $N_S$ (–) are Sips model parameters. The Sips model becomes the Langmuir model if $N_S = 1$.

The commercial software package TableCurve 2D® (Version 5.1, SYSTAT Software, Inc., Chicago, IL, USA) was used to estimate the single sorption model parameters.

*2.6. Binary Sorption Models*

To compare the single sorption with those of the multi-solute sorption systems, the P-factor model is used by introducing a "lumped" capacity factor, $P_i$ [28]:

$$P_i = \frac{q_{mL,i}}{q^*_{mL,i}} \tag{7}$$

where $q_{mL,i}$ and $q_{mL,i}^*$ are the monolayer capacity for component $i$ in the single and binary sorption system, respectively. By assuming this model as a Langmuir model for each component $i$, the binary sorption model equation is expressed as

$$q_i = \frac{1}{P_i} \frac{b_{L,i} q_{mL,i} C_i}{1 + b_{L,i} C_i} \qquad (8)$$

where $b_{L,i}$ and $q_{mL,i}$ can be estimated from single sorption.

To predict the sorbed phase of heavy metal concentrations using aqueous-phase concentrations, the IAST proposed by Radke and Prausnitz [29] can be used, where a material balance on each solute is required. The equivalence of the spreading pressure ($\pi$) in a mixture containing $N$ solutes leads to [30,31]:

$$\pi = \frac{RT}{A} \int_0^{q_1^*} \frac{d \log C_1}{d \log q_1} dq_1 = \frac{RT}{A} \int_0^{q_2^*} \frac{d \log C_2}{d \log q_2} dq_2 = \cdots = \frac{RT}{A} \int_0^{q_N^*} \frac{d \log C_N}{d \log q_N} dq_N \qquad (9)$$

or

$$\pi = \frac{RT}{A} \int_0^{C_1^*} \frac{q_1}{C_1} dC_1 = \frac{RT}{A} \int_0^{C_2^*} \frac{q_2}{C_2} dC_2 = \cdots = \frac{RT}{A} \int_0^{C_N^*} \frac{q_N}{C_N} dC_N \qquad (10)$$

where $R$ is the universal gas constant, $T$ is the temperature at which sorption occurs, and $A$ is the interfacial area between the solution and solid sorbent. The IAST calculations are explained in detail elsewhere [31,32].

For all single and binary sorption models, the coefficient of determination ($R^2$), the sum of squared errors (SSE), and the root mean square error (RMSE) were calculated, whereas for the binary sorption models, the RMSE calculation was also added to evaluate if the model fits experimental data [24].

The P-factor model was estimated using Microsoft Office Excel (Version 2016, Microsoft Company Ltd., Redmond, WA, USA).

## 3. Results and Discussion

### 3.1. Physicochemical Properties of IOCS and MOCS

The physicochemical properties of IOCS and MOCS such as $A_{BET}$, CEC, and $pH_{PZC}$ are summarized in Table 1.

**Table 1.** Physicochemical properties of the sorbents.

|  | IOCS | MOCS |
|---|---|---|
| BET surface area ($A_{BET}$, m$^2$ g$^{-1}$) | 0.711 | 1.057 |
| CEC (mmol 100 g$^{-1}$) | 1.389 | 2.670 |
| $pH_{PZC}$ | 8.2 | 7.1 |

The $A_{BET}$ and CEC of IOCS (0.711 m$^2$ g$^{-1}$, 1.389 mmol 100 g$^{-1}$) were lower than those of MOCS (1.057 m$^2$ g$^{-1}$, 2.670 mmol 100 g$^{-1}$). Meanwhile, the $pH_{PZC}$ of IOCS (8.2) was higher than that of MOCS (7.1).

The XRD spectra are shown in Figure 1. Figure 1a represents the uncoated sand, which was similar to that reported in the literature [33]. A strong peak at 26.7° for both IOCS (Figure 1b) and MOCS (Figure 1c) was observed, corresponding to the silica in the sand [19,33,34]. In Figure 1b, some weak residual peaks in the range of 50–61° belonging to the iron oxide [33–35] were detected. The residual peaks of manganese oxide [12,35] also appeared in Figure 1c in the range of 30–60° [36].

Figure 2a,b depict the SEM images of IOSC and MOCS, respectively. The morphology showed that the particle sizes of IOCS and MOCS were almost similar (0.5–1.5 mm) with an irregular shape. The EDS analysis results of IOCS and MOCS are shown in Figure 2c,d, respectively. The elemental

compositions of IOCS (Figure 2c) were identical to those reported in the previous studies [15,17], where Fe indicated the presence of iron oxide in the IOSC, and Mn indicated the presence of manganese oxide in the MOSC (Figure 2c) [36].

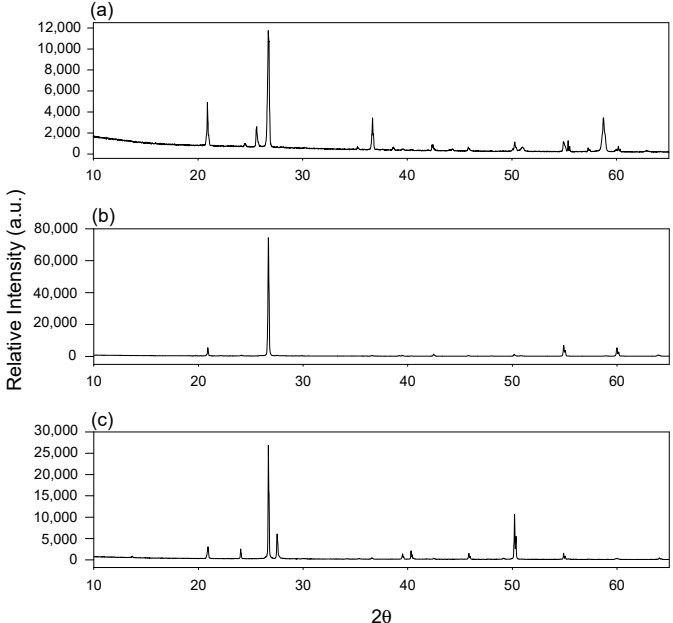

**Figure 1.** X-ray diffractometer (XRD) patterns of (**a**) uncoated sand, (**b**) iron oxide-coated sand (IOCS), and (**c**) manganese oxide-coated sand (MOCS).

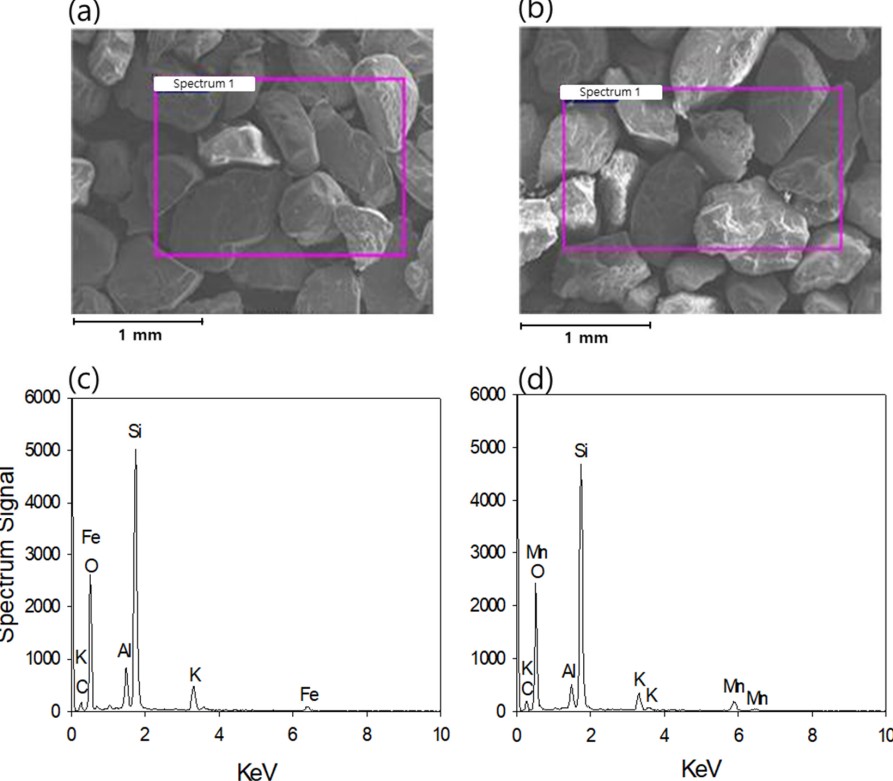

**Figure 2.** Physicochemical properties of sorbents: scanning electron microscopy (SEM) images of (**a**) iron oxide-coated sand (IOCS) and (**b**) manganese oxide-coated sand (MOCS); energy-dispersive X-ray spectrometer (EDS) peaks of (**c**) iron oxide-coated sand (IOCS) and (**d**) manganese oxide-coated sand (MOCS).

### 3.2. Single Sorption

Figure 3 shows the single sorption isotherms of Ni and Zn onto IOCS and MOCS at 0‰ and 30‰ salinity. The Freundlich, Langmuir, DR, and Sips model parameters are summarized in Table 2. The experimental data fitted all models well (Freundlich: $0.947 < R^2 < 0.989$, Langmuir: $0.989 < R^2 < 0.999$, DR: $0.969 < R^2 < 0.998$, and Sips: $0.992 < R^2 < 0.999$).

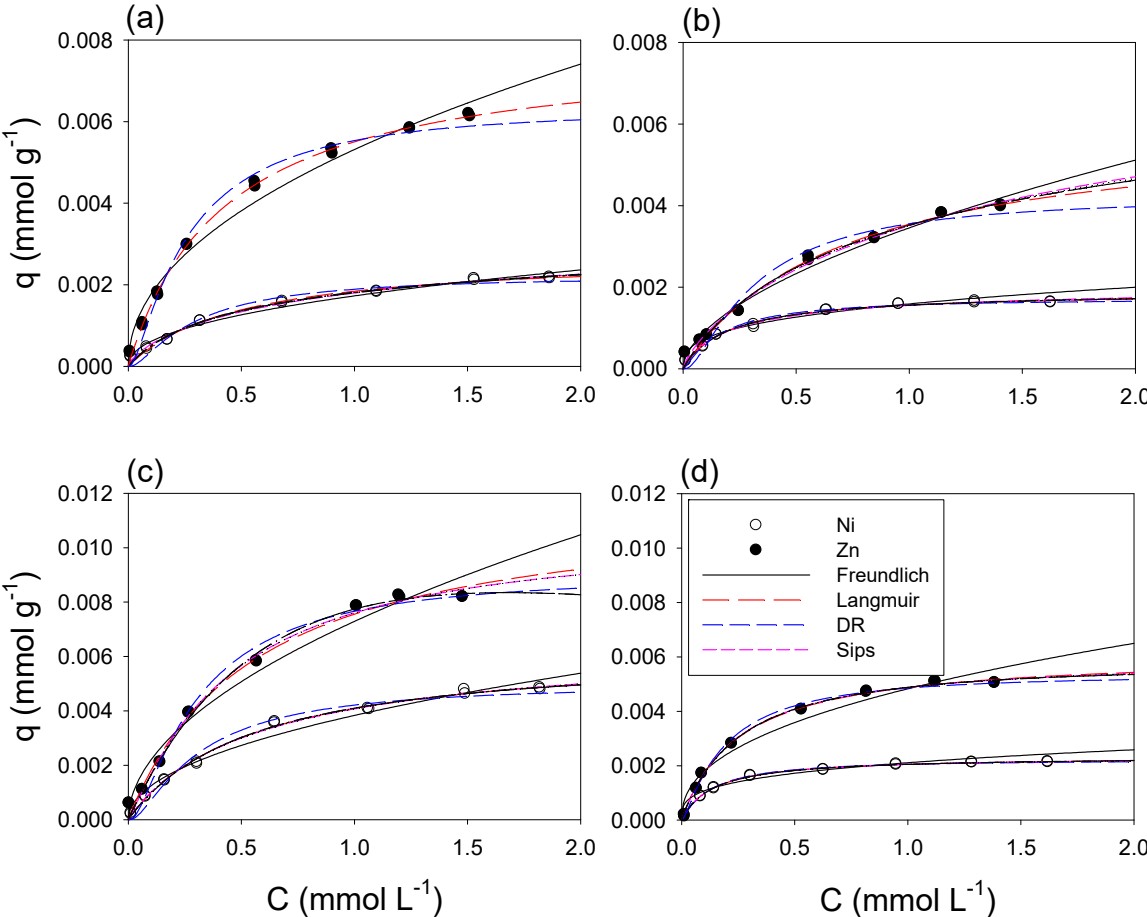

**Figure 3.** Single sorption of Ni and Zn onto iron oxide-coated sand (IOCS) at (**a**) 0‰ and (**b**) 30‰ salinity and onto manganese oxide-coated sand (MOCS) at (**c**) 0‰ and (**d**) 30‰ salinity. Lines represent the sorption model predictions.

**Table 2.** Freundlich, Langmuir, DR, and Sips model parameters for the single sorption of Ni and Zn onto IOCS and MOCS at pH = 5.

|  | Sorbent | Metal | Salinity | $K_F$ | $N_F$ (–) | $R^2$ | SSE |
|---|---|---|---|---|---|---|---|
| **Freundlich** | IOCS | Ni | 0 | 0.002 | 0.453 | 0.986 | 0.000 |
|  |  |  | 30 | 0.002 | 0.344 | 0.967 | 0.000 |
|  |  | Zn | 0 | 0.005 | 0.480 | 0.985 | 0.000 |
|  |  |  | 30 | 0.004 | 0.567 | 0.989 | 0.000 |
|  | MOCS | Ni | 0 | 0.004 | 0.490 | 0.982 | 0.000 |
|  |  |  | 30 | 0.002 | 0.309 | 0.947 | 0.000 |
|  |  | Zn | 0 | 0.007 | 0.523 | 0.978 | 0.000 |
|  |  |  | 30 | 0.005 | 0.429 | 0.966 | 0.000 |

**Table 2.** *Cont.*

| | Sorbent | Metal | Salinity | $q_{mL}$ | $b_L$ | | $R^2$ | SSE | |
|---|---|---|---|---|---|---|---|---|---|
| **Langmuir** | IOCS | Ni | 0 | 0.003 | 2.046 | | 0.989 | 0.000 | |
| | | | 30 | 0.002 | 4.919 | | 0.991 | 0.000 | |
| | | Zn | 0 | 0.008 | 2.324 | | 0.998 | 0.000 | |
| | | | 30 | 0.006 | 1.409 | | 0.990 | 0.000 | |
| | MOCS | Ni | 0 | 0.006 | 1.815 | | 0.996 | 0.000 | |
| | | | 30 | 0.002 | 7.199 | | 0.999 | 0.000 | |
| | | Zn | 0 | 0.012 | 1.779 | | 0.993 | 0.000 | |
| | | | 30 | 0.006 | 3.905 | | 0.997 | 0.000 | |
| | **Sorbent** | **Metal** | **Salinity** | $q_{mD}$ | β | | $R^2$ | SSE | *E* |
| **DR** | IOCS | Ni | 0 | 0.002 | 5.177 | | 0.989 | 0.000 | 3.11 |
| | | | 30 | 0.002 | 2.913 | | 0.984 | 0.000 | 4.14 |
| | | Zn | 0 | 0.006 | 4.545 | | 0.989 | 0.000 | 3.32 |
| | | | 30 | 0.004 | 5.713 | | 0.969 | 0.000 | 2.96 |
| | MOCS | Ni | 0 | 0.005 | 5.330 | | 0.983 | 0.000 | 3.01 |
| | | | 30 | 0.002 | 2.292 | | 0.998 | 0.000 | 4.67 |
| | | Zn | 0 | 0.009 | 5.444 | | 0.988 | 0.000 | 3.03 |
| | | | 30 | 0.005 | 3.238 | | 0.985 | 0.000 | 3.93 |
| | **Sorbent** | **Metal** | **Salinity** | $q_{mS}$ | $b_S$ | $N_S$ (–) | $R^2$ | SSE | |
| **Sips** | IOCS | Ni | 0 | 0.004 | 0.975 | 0.752 | 0.992 | 0.000 | |
| | | | 30 | 0.002 | 3.896 | 0.831 | 0.993 | 0.000 | |
| | | Zn | 0 | 0.008 | 2.057 | 0.941 | 0.999 | 0.000 | |
| | | | 30 | 0.009 | 0.589 | 0.793 | 0.992 | 0.000 | |
| | MOCS | Ni | 0 | 0.007 | 1.661 | 0.957 | 0.996 | 0.000 | |
| | | | 30 | 0.002 | 7.283 | 1.016 | 0.999 | 0.000 | |
| | | Zn | 0 | 0.011 | 2.230 | 1.131 | 0.993 | 0.000 | |
| | | | 30 | 0.006 | 4.155 | 1.043 | 0.998 | 0.000 | |

Unit: Salinity (‰), $K_F$ [(mmol g$^{-1}$)/(mmol L$^{-1}$) $N_F$], $q_{mL} = q_{mD} = q_{mS}$ (mmol g$^{-1}$), $b_L = b_S$ (L mmol$^{-1}$), β (mol$^2$ J$^{-2}$, × 10$^{-8}$), *E* (kJ mol$^{-1}$). Calculated at $C_0 = 0.02$ mM.

Based on the $N_F$ values of the Freundlich model (0.309–0.567; Table 2), all sorption isotherms were nonlinear, indicating the heterogeneity of the IOCS and MOCS surface. The $K_F$ values for Ni and Zn at 0‰ salinity were steadily higher than those at 30‰ salinity for both IOCS and MOCS, indicating that the salinity affected the sorption affinity by decreasing the $K_F$ values of Ni and Zn onto IOCS and MOCS. On the other hand, the $K_F$ values for IOCS were slightly lower than those for MOCS.

By assuming a finite number of active sites on the IOCS and MOCS, the single sorption data fitted the Langmuir model. The $q_{mL}$ values from the Langmuir model for MOCS were higher than those for IOCS for both Ni and Zn at 0‰ and 30‰ salinity (Table 2), indicating that the sorption capacity of MOCS was higher than that of IOCS due to the higher $A_{BET}$ and CEC of MOCS than those of IOCS (Table 1).

The $q_{mL}$ values of Ni onto IOCS (0.002–0.008 mmol g$^{-1}$) and MOCS (0.002–0.012 mmol g$^{-1}$) used in this study were slightly lower than those reported in the previous studies (0.007–0.017 mmol g$^{-1}$) [17,19]. The comparison of sorption capacities of various heavy metal onto IOCS and MOCS in water are summarized in Table S2. The decrease in the $q_{mL}$ values of IOCS and MOCS as the salinity changed from 0 to 30‰ indicates that the salinity affected their sorption capacities. In terms of sorption favorability, the Ni and Zn sorption onto both IOCS and MOCS at both 0‰ and 30‰ salinity were favorable, as shown by the $R_L$ values (0 < $R_L$ < 1; Figure S2).

Similar to the $q_{mL}$ values of the Langmuir model, the $q_{mD}$ values according to the DR model decreased as the salinity changed from 0 to 30‰ (Table 2), indicating that the salinity also affected the $q_{mD}$ values. The physical process was the primary mechanism of Ni and Zn sorption onto both IOCS and MOCS at both 0‰ and 30‰ salinity, as the calculated $E$ values were <8 kJ mol$^{-1}$. The $q_{mS}$ values of the Sips model also decreased when the salinity changed from 0 to 30‰ (Table 2), suggesting that the salinity also affected the $q_{mS}$ values.

### 3.3. Binary Sorption

In this study, the Langmuir model was used to predict the binary sorption data by calculating the ratio of the sorption capacity of the corresponding metal ion in the single sorption system ($q_{mL}$) to the sorption capacity of one metal ion in the binary sorption system ($q_{mL}^*$). Then, the competition effect can be interpreted as follows: if the ratio is less than 1, the sorption is enhanced by the presence of the other metal ion; if the ratio is equal to 1, no competition effect is observed; if the ratio is greater than 1, the competition is preceded [25].

Similarly, the ratio calculation can be applied to the site energy factor in the single sorption system ($b_L$) to that in the binary sorption system ($b_L^*$) [24]. As summarized in Table 3, the binary sorption of Ni/Zn onto IOCS and MOCS at 0‰ and 30‰ salinity was fitted to the Langmuir model, where the competition occurred as indicated by the higher $q_{mL}^*$ and $b_L^*$ values than the $q_{mL}$ and $b_L$ values.

**Table 3.** Langmuir model parameters for the binary sorption (competitive adsorption) of Ni and Zn onto IOCS and MOCS at pH = 5.

| Sorbent | Salinity (‰) | Solute | $q_{mL}^*$ | $b_L^*$ | $R^2$ | SSE |
|---------|--------------|--------|-----------|---------|-------|-----|
| IOCS | 0 | Ni in Ni/Zn | 0.004 | 2.000 | 0.900 | 0.000 |
| | | Zn in Ni/Zn | 0.007 | 3.234 | 0.979 | 0.000 |
| | 30 | Ni in Ni/Zn | 0.003 | 3.425 | 0.944 | 0.000 |
| | | Zn in Ni/Zn | 0.006 | 3.927 | 0.984 | 0.000 |
| MOCS | 0 | Ni in Ni/Zn | 0.005 | 1.575 | 0.973 | 0.000 |
| | | Zn in Ni/Zn | 0.008 | 3.287 | 0.988 | 0.000 |
| | 30 | Ni in Ni/Zn | 0.003 | 3.064 | 0.946 | 0.000 |
| | | Zn in Ni/Zn | 0.006 | 3.899 | 0.984 | 0.000 |

Unit: $q_{mL}^*$ (mmol g$^{-1}$), $b_L^*$ (L mmol$^{-1}$). $q_{mL}^*$ and $b_L^*$ indicate the $q_{mL}$ value and $b_L$ value for binary sorption, respectively.

The $q_{mL,Ni}/q_{mL,Zn}$ ratios onto IOCS and MOCS at both 0‰ and 30‰ salinity were less than 1 (Table 4). The same holds for $q_{mL,Ni}^* / q_{mL,Zn}^*$. These ratios indicated that the sorption capacity of Zn in the Ni/Zn system onto IOCS and MOCS at both 0‰ and 30‰ salinity in the single and binary sorption systems was more dominant than that of Ni due to the higher $q_{mL,Zn}$ and $q_{mL,Zn}^*$ than the $q_{mL,Ni}$ and $q_{mL,Ni}^*$. The increase of selectivity of IOCS and MOCS towards Zn in the presence of Ni was responsible for the increase of Zn sorption in the Ni/Zn binary system. A similar synergistic effect of Zn was also reported by Chen et al. [37] explaining the increase of Zn sorption in the multi-solute system of Zn/Pb/Cu due to the increase of Zn selectivity onto TiO$_2$ composite. The $b_{L,Ni}/b_{L,Zn}$ ratios were generally greater than 1, except for IOCS at 0‰ salinity, whereas the $b_{L,Ni}^*/b_{L,Zn}^*$ ratios were all less than 1. This result indicated that the Ni had higher affinity than Zn to the sorption sites of both IOCS and MOCS in the single sorption system, whereas the Ni affinity was lower than Zn in the binary sorption system regardless of salinity. The $q_{mL,Ni}/q_{mL,Ni}^*$ ratios were greater than 1, except for MOCS at 0‰ salinity, while all $q_{mL,Zn}/q_{mL,Zn}^*$ ratios were less than 1. This result suggests that the Ni sorption capacity in the single sorption was higher than those in the binary sorption due to the competition, whereas the Zn sorption capacity in the single sorption was lower than those in the binary sorption onto IOCS and MOCS at both 0‰ and 30‰ salinity.

**Table 4.** Comparison of $q_{mL}$ and $b_L$ values of single and binary sorption of Ni and Zn at pH = 5.

| Sorbent | Salinity (‰) | $q_{mL,Ni}/q_{mL,Zn}$ | $q^*_{mL,Ni}/q^*_{mL,Zn}$ | $q_{mL,Ni}/q^*_{mL,Ni}$ | $q_{mL,Zn}/q^*_{mL,Zn}$ |
|---|---|---|---|---|---|
| IOCS | 0 | 0.348 | 0.539 | 1.449 | 0.935 |
|  | 30 | 0.310 | 0.450 | 1.344 | 0.926 |
| MOCS | 0 | 0.537 | 0.471 | 0.419 | 0.477 |
|  | 30 | 0.381 | 0.471 | 1.138 | 0.919 |
| **Sorbent** | **Salinity (‰)** | $b_{L,Ni}/b_{L,Zn}$ | $b^*_{L,Ni}/b^*_{L,Zn}$ | $b_{L,Ni}/b^*_{L,Ni}$ | $b_{L,Zn}/b^*_{L,Zn}$ |
| IOCS | 0 | 0.880 | 0.618 | 0.978 | 1.392 |
|  | 30 | 3.491 | 0.872 | 0.696 | 2.787 |
| MOCS | 0 | 1.021 | 0.786 | 1.688 | 2.192 |
|  | 30 | 1.843 | 0.786 | 0.426 | 0.998 |

$q_{mL,Ni}$ and $q_{mL,Zn}$ indicate $q_{mL}$ values for single sorption, and $q^*_{mL,Ni}$ and $q^*_{mL,Zn}$ indicate the $q_{mL}$ values for binary sorption, respectively. $b_{L,Ni}$ and $b_{L,Zn}$ indicate $b_L$ values for single sorption, and $b^*_{L,Ni}$ and $b^*_{L,Zn}$ indicate $b_L$ values for binary sorption (competitive adsorption), respectively.

The experimental data of the binary sorption of Ni and Zn onto IOCS and MOCS fitted using the P-factor model are shown in Figure 4. The model parameters are summarized in Table 5. The P-factor model predicted the binary sorption data well as shown in Table 5 ($0.790 < R^2 < 0.974$). The $P_i$ values of Ni and Zn sorptions onto MOCS decreased remarkably as the salinity increased from 0 to 30‰, whereas the $P_i$ values of Ni and Zn sorptions onto IOCS slightly changed. This result suggests that the decrease in the monolayer sorption capacity of MOCS was more pronounced than that of IOCS. In the case of Zn, the $P_i$ values were all more than 1, confirming that the Zn sorptions were hindered by the presence of Ni. Meanwhile, all the $P_i$ values of Ni, except for the MOCS at 0‰, were less than 1, indicating less hindrance of Ni sorption by the presence of Zn. A similar tendency was also reported by Sdiri et al. [38], who reported the higher hindrance of Zn sorptions in the presence of Cu(II) in the binary system of Cu(II)/Zn due to the strong inhibitory effect, where the $P_i$ values of Zn were much higher than 1.

Further, the binary sorption data were also predicted using the IAST model coupled with the single sorption model of Freundlich (IAST–Freundlich), Langmuir (IAST–Langmuir), and Sips (–) (Table 5 and Figure 4). The Freundlich, Langmuir, and Sips model parameters (Table 2) were used for the fitting process. The IAST–Freundlich, IAST–Langmuir, and IAST–Sips models were satisfactory in predicting the Zn sorption in the Ni/Zn system for IOCS and MOCS at both 0‰ and 30‰ salinity ($0.808 < R^2 < 0.988$), whereas the predictions for Ni sorption were relatively poor ($0.247 < R^2 < 0.860$). The fitting result of Zn was in good agreement with the IAST main assumption of ideal solution [29], whereas the interaction between the Ni and heterogeneous sorption site of IOCS and MOCS was responsible for the Ni poor fitting result [39].

Overall, the IOCS and MOCS are promising reactive media for PRB because they were satisfactory in removing Ni and Zn from groundwater via physical sorption. In comparison to the widely used PRB reactive media, such as apatite [24,40], the sand used for metal oxide-solid support is abundant, and the preparations of IOCS and MOCS [12,19] are relatively simpler and require less chemicals than that of apatite [25]. Although the sorption capacities were affected by the salinity, the IOCS and MOCS were still effective for Ni and Zn sorption in the binary system, where the Zn sorption capacities were higher in the binary system than that in the single system. This result is promising for PRB application, especially for removing Ni and Zn from groundwater in the influence of salinity. Moreover, the sorption properties of Ni and Zn in the single and binary systems in groundwater in the presence of salinity can be well predicted by using single and binary sorption models. Therefore, the single and binary sorption models are useful for predicting the Ni and Zn sorption onto PRB.

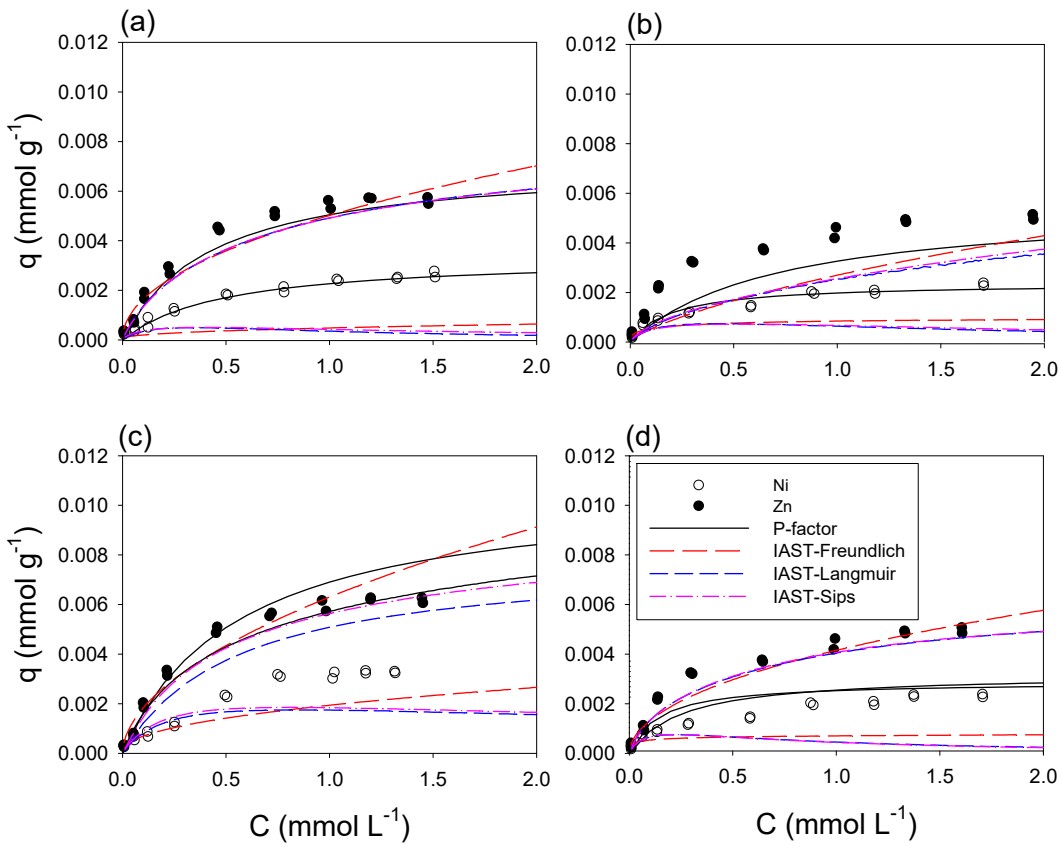

**Figure 4.** Competitive sorption of Ni and Zn onto iron oxide-coated sand (IOCS) at (**a**) 0‰ and (**b**) 30‰ salinity and onto manganese oxide-coated sand (MOCS) at (**c**) 0‰ and (**d**) 30‰ salinity. Lines represent the binary sorption model prediction.

**Table 5.** Model parameters for the binary sorption of Ni(1) and Zn(2) onto IOCS and MOCS at pH = 5.

| Sorbent | Salinity (‰) | Sorption Model | $P_i$ | $R^2$ | SSE | RMSE |
|---|---|---|---|---|---|---|
| IOCS | 0 | P-factor<br>IAST–Freundlich<br>IAST–Langmuir<br>IAST–Sips | 0.690/1.069 | 0.967/0.972<br>0.294/0.960<br>0.247/0.954<br>0.284/0.954 | 0.000/0.000<br>0.000/0.000<br>0.000/0.000<br>0.000/0.000 | 0.000/0.001<br>0.002/0.001<br>0.002/0.001<br>0.002/0.001 |
| | 30 | P-factor<br>IAST–Freundlich<br>IAST–Langmuir<br>IAST–Sips | 0.744/1.080 | 0.974/0.911<br>0.699/0.837<br>0.556/0.808<br>0.575/0.823 | 0.000/0.000<br>0.000/0.000<br>0.000/0.000<br>0.000/0.000 | 0.000/0.001<br>0.001/0.002<br>0.001/0.002<br>0.001/0.002 |
| MOCS | 0 | P-factor<br>IAST–Freundlich<br>IAST–Langmuir<br>IAST–Sips | 1.243/1.532 | 0.790/0.820<br>0.860/0.973<br>0.809/0.988<br>0.831/0.986 | 0.000/0.000<br>0.000/0.000<br>0.000/0.000<br>0.000/0.000 | 0.001/0.002<br>0.001/0.001<br>0.001/0.000<br>0.001/0.000 |
| | 30 | P-factor<br>IAST–Freundlich<br>IAST–Langmuir<br>IAST–Sips | 0.877/1.088 | 0.931/0.811<br>0.623/0.980<br>0.454/0.986<br>0.447/0.986 | 0.000/0.000<br>0.000/0.000<br>0.000/0.000<br>0.000/0.000 | 0.000/0.002<br>0.001/0.000<br>0.001/0.000<br>0.001/0.000 |

## 4. Conclusions

The influence of salinity on the single and binary sorption of Ni and Zn onto IOCS and MOCS was investigated at pH 5. IOCS and MOCS are good sorbents for Ni and Zn. The sorption capacity of MOCS was higher than that of IOCS. Based on the nonlinearity of the sorption isotherms,

the Freundlich, Langmuir, DR, and Sips models accurately predicted the single sorption experimental data. The Ni sorption capacities in the single sorption system were higher than those in the binary sorption system, whereas the Zn sorption capacities in the single sorption system were lower than those in the binary sorption for both IOCS and MOCS at both 0‰ and 30‰ salinity. The salinity reduced Ni and Zn sorption onto IOCS and MOCS in both single and binary sorption systems. Satisfactory predictions were achieved by using binary sorption models, including P-factor, IAST–Freundlich, IAST–Langmuir, and IAST–Sips. Among these, the P-factor showed the best fitting results in predicting the influence of salinity on Ni and Zn binary sorption onto IOCS and MOCS. These results show that these models can be utilized to study the binary sorption of various heavy metals onto metal oxide-coated sand. IOCS and MOCS can be applied as a replaceable reactive media in the PRB application for Ni and Zn in the presence of salinity. The use of IOCS and MOCS in the PRB application can be combined with more reactive media to improve their sorption capacities.

**Supplementary Materials:** The following are available online at http://www.mdpi.com/2071-1050/12/14/5815/s1, Figure S1: Total concentration (%) of $Ni^{2+}$ at (a) 0‰ and (b) 30‰ salinity and $Zn^{2+}$ at (c) 0‰, and (d) 30‰ salinity as a function of pH predicted by MINEQL+ for Windows (version 4.6, Environmental Research Software, USA). A 0.02 mM $Ni^{2+}$ or $Zn^{2+}$ solution was used for the prediction, Figure S2: RL value calculations of Langmuir isotherm for single sorption of Ni and Zn onto iron oxide-coated sand (IOCS) at (a) 0‰ and (b) 30‰ salinity and onto manganese oxide-coated sand (MOCS) at (c) 0‰ and (d) 30‰ salinity, Table S1: The chemical compositions of artificial seawater (30‰), Table S2: Sorption capacities of various heavy metal onto IOCS and MOCS in water.

**Author Contributions:** Conceptualization, J.C., A.S. and W.S.S.; methodology, J.C., A.S. and W.S.S.; software, W.S.S.; validation, W.S.S.; investigation, J.C. and A.S.; data curation, J.C., A.S. and W.S.S.; writing—original draft preparation, A.S.; review and editing, W.S.S.; supervision, W.S.S.; project administration, W.S.S.; funding acquisition, W.S.S. All authors have read and agreed to the published version of the manuscript.

**Funding:** This research received no external funding.

**Acknowledgments:** This work was supported by Korea Environment Industry and Technology Institute (KEITI) through The Chemical Accident Prevention Technology Development Project, funded by Korea Ministry of Environment (MOE) (2019001960002).

**Conflicts of Interest:** The authors declare no conflict of interest.

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
