# Peer review of "The Influence of Salinity on the Removal of Ni and Zn by Sorption onto Iron Oxide- and Manganese Oxide-Coated Sand"

_sustainability, doi:10.3390/su12145815_

Round 1

Reviewer 1 Report

The submitted ms “The Influence of Salinity on the Removal of Ni and Zn by Sorption onto Iron Oxide- and Manganese Oxide-Coated Sand” is focused on the adsorption of Zn and Ni from single and binary solutions by IOCS and MOCS at different salinity.

Comments:

  1. In Material and methods section the procedure regarding the solution salinity has to be included.
  2. I am surprised by very low adsorption capacity of both sorbents (3 – 12 µmol/g determined from Langmuir model). In my opinion, such low values limit the practical application of your modified sands PRB.
  3. In lines 295 – 298 authors declare that Ni adsorption in both single and binary systems is dominant. However, in Table 4 higher values of Qmax and b for Zn (from Langmuir model) in both single and binary systems are stated indicating the higher affinity of sorbents to Zn not Ni. Please clarify and improve discussion. To investigate competitive effects in binary system, 3D adsorption surfaces are very useful tool, see e.g. Journal of Radioanalytical and Nuclear Chemistry (2019) 319, 855-867.
  4. Authors discuss competition only in binary Ni-Zn system. However, due to increasing salinity increase the concentration of competing cations and competition occurs also in single systems.
  5. Regarding the SRS and MA models, the values of competition coefficients α12 and α21 ranging from zero (no competition) to values greater than zero (no lower!). Please carefully consider calculations used for parameter determination and interpretation (https://doi.org/10.1016/0169-7722(93)90027-P).
  6. The overall RandD part lack discussion of obtained results.

Author Response

General comments:

The submitted ms “The Influence of Salinity on the Removal of Ni and Zn by Sorption onto Iron Oxide- and Manganese Oxide-Coated Sand” is focused on the adsorption of Zn and Ni from single and binary solutions by IOCS and MOCS at different salinity.

Specific comments or questions:
Point 1: In Material and methods section the procedure regarding the solution salinity has to be included.

Response 1: The preparation of artificial seawater (salinity) was rewritten in detail in the Materials and Methods section:

“Artificial seawater was prepared according to a standard method (US EPA 2002). The artificial seawater consisted of Na+, K+, Ca2+, Mg2+, Cl, SO42, and HCO3, was prepared by dissolving NaCl, KCl, CaCl2, MgCl2, MgSO4, and NaHCO3 in 1 L of DDI water, as shown in Supplementary Materials (SI) Table S1.” (P 2, L 78–80).

“Series of Ni (0.02–1.70 mmol L1) and Zn solution (0.02–1.53 mmol L1) were prepared using DDI water and artificial seawater to investigate the influence of salinity on the sorption.” (P 3, L 115–117).

Reference:

United States Environmental Protection Agency (US EPA). Methods for measuring the acute toxicity of effluents and receiving waters to freshwater and marine organisms. Washington DC, 2002, EPA-821-R-02-012.

Point 2: I am surprised by very low adsorption capacity of both sorbents (3 – 12 µmol/g determined from Langmuir model). In my opinion, such low values limit the practical application of your modified sands PRB.

Response 2: Please note that this is the preliminary study of Ni and Zn sorption onto IOCS and MOCS in the presence of salinity in order to investigate the capability of IOCS and MOCS as PRB material. The sorption capacity of IOCS and MOCS for Ni and Zn removal in the PRB application can be improved by combining the IOCS and MOCS with more cost-efficient and higher sorption capacity of natural sorbents such as fishbone or apatite. We are currently submitting a paper on the Ni and Zn sorption onto fishbone and hydroxyapatite.

Point 3: In lines 295 – 298 authors declare that Ni adsorption in both single and binary systems is dominant. However, in Table 4 higher values of Qmax and b for Zn (from Langmuir model) in both single and binary systems are stated indicating the higher affinity of sorbents to Zn not Ni. Please clarify and improve discussion. To investigate competitive effects in binary system, 3D adsorption surfaces are very useful tool, see e.g. Journal of Radioanalytical and Nuclear Chemistry (2019) 319, 855-867.

Response 3: As the reviewer commented, the discussion was revised based on the comparison of qmL and bL values of single and binary sorption in Table 4:

“These ratios indicated that the sorption capacity of Zn in the Ni/Zn system onto IOCS and MOCS at both 0‰ and 30‰ salinity in the single and binary sorption systems was more dominant than that of Ni due to the higher qmL,Zn and  than the qmL,Ni and .” (P 10, L 288–291).

- We also tried to present the binary sorption data using 3D Figures. However, the final Figures were visually not clear. Therefore, we used 2D figures to present the binary sorption data more clearly.

Point 4: Authors discuss competition only in binary Ni-Zn system. However, due to increasing salinity increase the concentration of competing cations and competition occurs also in single systems.

Response 4: As the reviewer mentioned, we also considered the competition between Ni or Zn and the cations and anions consisted in the artificial seawater (i.e., Na+, K+, Ca2+, Mg2+, Cl, SO42, and HCO3). Therefore, we predicted metal speciation of Ni and Zn in the presence of cations and anions in the artificial seawater using a software MINEQL+ for Windows (version 4.6, Environmental Research Software, USA). As shown in Figure S1 (Supplementary Materials), the dissolved Ni and Zn were still dominant (Figures S1b and d), indicating that the dissolved Ni and Zn were not highly affected by the cations and anions competitions in the artificial seawater.

Point 5: Regarding the SRS and MA models, the values of competition coefficients α12 and α21 ranging from zero (no competition) to values greater than zero (no lower!). Please carefully consider calculations used for parameter determination and interpretation (https://doi.org/10.1016/0169-7722(93)90027-P)

Response 5: The MA and SRS model fitting were deleted in the revised manuscript to avoid confusion.  

Point 6: The overall Results and Discussion part lack discussion of obtained results.

Response 6: The Result and Discussion section were intensively revised and rewritten considering the reviewer’s comments and suggestion.

Reviewer 2 Report

The manuscript represents a typical physical and chemical investigation on metal sorption and its utility has a practical value for pollution prevention control, remediation and heavy metals removal from subsurface groundwaters. It hasn’t significant flaws, but needs in minor revision

Some remarks.

Please, pay attention on the equation (3). According to it RL cann’t be above 1, as stated in the text.

Please, correct parameter Ni in the equation (7). It must be NFi.

Figure 5 is a copy of figure 4. It mast be removed.

Author Response

General comments:

The manuscript represents a typical physical and chemical investigation on metal sorption and its utility has a practical value for pollution prevention control, remediation and heavy metals removal from subsurface groundwaters. It hasn’t significant flaws, but needs in minor revision.

Specific comments or questions:
Point 1: Please, pay attention on the equation (3). According to it RL can’t be above 1, as stated in the text.

Response 1: The RL definition was corrected:

“The RL values (–) indicate the type of isotherm to be favorable (0 < RL < 1), or irreversible (RL = 0)” (P 4, L 150–151).

Point 2: Please, correct parameter Niin the equation (7). It must be NFi.

Response 2: The equation (7) (MA model) was removed in the revised manuscript.

Point 3: Figure 5 is a copy of figure 4. It must be removed.

Response 3: Figures 4 and 5 in the previous manuscript were combined in Figure 4 in the revised manuscript (P 11, L 306).

Reviewer 3 Report

In this manuscript, the authors investigated the single/binary sorption capacity of Ni and Zn onto IOCS and MOCS. The first, the author prepared IOCS and MOCS as sorbents and then studied single and binary sorption mechanisms. The authors applied several classic models in the field and compared sorption capacity. Finally provided the comparison between two sorbents. Overall, this study fits the aim and scope of sustainability and could give good guidance in the field of groundwater treatment.

The specific comments are:

  1. This experiment was conducted when the temperature was at 25 °C, however, in groundwater, it is hard to reach that high temperature. So, what’s the reason of this selected temperature? Will the cold temperature in the real groundwater environment affect the results?
  2. The authors gave the reason why choosing pH 5 in this study (Line 114-116), however, in real groundwater environment, it is hard to main this pH, so are IOCS and MOCS still applicable when pH is not at 5?
  3. Line 123, please use the unit of “g” instead of “rpm”.
  4. Line 207, which version of MATLAB do you use?
  5. Figures 3 and 4, please consider change colors or forms. Some model fitting curves were presented dotted line, and they are too close.
  6. Figure 5, two curves use the same lines, very confusing.

Author Response

General comments:

In this manuscript, the authors investigated the single/binary sorption capacity of Ni and Zn onto IOCS and MOCS. The first, the author prepared IOCS and MOCS as sorbents and then studied single and binary sorption mechanisms. The authors applied several classic models in the field and compared sorption capacity. Finally provided the comparison between two sorbents. Overall, this study fits the aim and scope of sustainability and could give good guidance in the field of groundwater treatment.

Specific comments or questions:
Point 1: This experiment was conducted when the temperature was at 25 °C, however, in groundwater, it is hard to reach that high temperature. So, what’s the reason of this selected temperature? Will the cold temperature in the real groundwater environment affect the results?

Response 1: Please note that this is the preliminary study of Ni and Zn sorption onto IOCS and MOCS in the presence of salinity, where the experiments were conducted in the ideal temperature condition (i.e., 25 °C). Unfortunately, the effect of temperature on the sorption was not fully investigated at this stage. However, this is subjected to further investigation.

Point 2: The authors gave the reason why choosing pH 5 in this study (Line 114-116), however, in real groundwater environment, it is hard to maintain this pH, so are IOCS and MOCS still applicable when pH is not at 5?

Response 2: The Ni and Zn sorptions onto IOCS and MOCS were conducted at pH (i.e., pH 5.0). At pH 5.0, both Zn and Ni exist as ions. At pH > 5.0, the Ni and Zn precipitations start to occur. As explained in the section 2.4 Sorption Experiments (P 3, L 118–119), as well as illustrated in Figure S1 by MINEQL+, the dissolved Ni and Zn were still dominant until the pH 5.5 for both at 0 and 30‰ salinity. Above pH 5.5, the Ni and Zn start to precipitate, therefore pH 5.0 was chosen to prevent precipitates formation.

Point 3: Line 123, please use the unit of “g” instead of “rpm”

Response 3: The unit of “rpm” was converter to “g”; 3,000 rpm = 1,000 g (P 3, L 124).

Point 4: Line 207, which version of MATLAB do you use?

Response 4: Since the MA and SRS models were removed in the revision, the explanation on the Matlab curve-fitting toolbox for MA and SRS model fitting was also deleted.

Point 5: Figures 3 and 4, please consider change colors or forms. Some model fitting curves were presented dotted line, and they are too close.

Response 5: The line colors in Figures 3 (P 8, L 245) and 4 (P 11, L 306) were changed.

Point 6: Figure 5, two curves use the same lines, very confusing

Response 6: Figures 4 and 5 in the previous manuscript were combined in Figure 4 in the revised manuscript (P 11, L 306).

Round 2

Reviewer 1 Report

The submitted revised version of ms “The Influence of Salinity on the Removal of Ni and Zn by Sorption onto Iron Oxide- and Manganese Oxide-Coated Sand” is focused on the adsorption of Zn and Ni from single and binary solutions by IOCS and MOCS at different salinity. Although some minor corrections have been made I still feel that the work is not well presented and contains several shortcomings.

  1. The crystal sizes of IOCS and MOCS calculated using Scherrer equation were 0.6 and 0.7 mm! However, this equation it does not apply to grains larger than about 0.1 to 0.2 μm.
  2. How do you explain the higher Zn adsorption in the binary system than in a single system? It does not make sense.
  3. Results from binary data fitting are poorly discussed and compared with other authors. In my previous review, I highlight this problem, however, I can not find any serious discussion in revised ms.
  4. In conclusion authors stated that IOCS and MOCS are sustainable reactive media for Ni and Zn adsorption, however, the maximal adsorption capacities of studied materials are significantly lower than many materials used in adsorption studies. Improve conclusions.

Reviewer 3 Report

The authors have provided a good revise, and I agree to accept this paper. 

Author Response

Response to the comments by reviewer #3:

General comments:

The authors have provided a good revise, and I agree to accept this paper

Round 3

Reviewer 1 Report

Revised ms was improved and comments accepted. I advice to authors to include Table from "Response to comments from reviewer 1" at least to supplementary material. 

One final note to authors: not all preliminary experiments need to be published. Sometimes they are just a good basis for quality research.
